# Results of Numerical Modeling of Blood Flow in the Internal Jugular Vein Exhibiting Different Types of Strictures

**DOI:** 10.3390/diagnostics12112862

**Published:** 2022-11-18

**Authors:** Anas Rashid, Syed Atif Iqrar, Aiman Rashid, Marian Simka

**Affiliations:** 1Department of Neuroscience “Rita Levi Montalcini”, University of Torino, 10125 Torino, Italy; 2Department of Electronics and Telecommunications, Polytechnic of Torino, 10129 Torino, Italy; 3Aston Institute of Photonic Technologies, College of Engineering and Physical Sciences, Aston University, Birmingham B4 7ET, UK; 4Department of Electrical and Electronic Engineering, University of Cagliari, 09123 Cagliari, Italy; 5Department of Anatomy, University of Opole, 45-052 Opole, Poland

**Keywords:** computational fluid dynamics, flow separation, internal jugular vein, numerical modeling, fluid–structure interaction

## Abstract

The clinical relevance of nozzle-like strictures in upper parts of the internal jugular veins remains unclear. This study was aimed at understanding flow disturbances caused by such stenoses. Computational fluid dynamics software, COMSOL Multiphysics, was used. Two-dimensional computational domain involved stenosis at the beginning of modeled veins, and a flexible valve downstream. The material of the venous valve was considered to be hyperelastic. In the vein models with symmetric 2-leaflets valve without upstream stenosis or with minor 30% stenosis, the flow was undisturbed. In the case of major 60% and 75% upstream stenosis, centerline velocity was positioned asymmetrically, and areas of reverse flow and flow separation developed. In the 2-leaflet models with major stenosis, vortices evoking flow asymmetry were present for the entire course of the model, while the valve leaflets were distorted by asymmetric flow. Our computational fluid dynamics modeling suggests that an impaired outflow from the brain through the internal jugular veins is likely to be primarily caused by pathological strictures in their upper parts. In addition, the jugular valve pathology can be exacerbated by strictures located in the upper segments of these veins.

## 1. Introduction

The internal jugular vein (IJV) is a paired blood vessel which constitutes the primary outflow route of blood from the brain. A number of studies suggested that impaired outflow through these veins can contribute to several neurodegenerative and neuroinflammatory disorders, like multiple sclerosis, Parkinson’s disease, lateral amyotrophic sclerosis, and Ménière disease [1,2,3,4,5]. The IJV begins intracranially, leaves the cranial cavity through the jugular foramen (opening in the temporal bone), and continues downward through the neck to join the brachiocephalic vein located in the upper chest (Figure 1). Normally, there are no strictures of IJV at the level of the jugular foramen. Still, some individuals present with stenosis of IJV in this area, usually caused by abnormal bony processes [6,7,8]. The clinical relevance of these stenoses remains unclear. Previous research on flow disturbances in IJV mostly focused on the jugular valve, which is the valve situated at the caudal end of IJV, just above its connection with the brachiocephalic vein [9,10,11,12].

It is worth highlighting the potential clinical relevance of our in silico study. As mentioned earlier, abnormal outflow through IJVs can be found in many patients with neurodegenerative and neuroinflammatory disorders. Currently, it is suspected that these neurological pathologies are associated with abnormal functioning of the astroglial-mediated interstitial fluid bulk flow, the so-called glymphatic system [13]. The activation of the glymphatic system primarily depends on a temporary decrease of the cortical blood flow, followed by a wave of inflow of the cerebrospinal fluid from the spinal canal to the cranial cavity [14]. Considering the Kellie–Monroe doctrine, proper interplay between the arterial and cerebrospinal flow requires an undisturbed venous outflow from the cranial cavity, and indeed an anomalous flow of the cerebrospinal fluid, as well as an imbalance between cerebral arterial and venous flow, has been demonstrated in multiple sclerosis patients [15,16,17]. About ten years ago, it was hoped that venous angioplasty for abnormal IJVs would be a much-awaited treatment for multiple sclerosis. However, a majority of randomized clinical trials on endovascular treatment for chronic cerebrospinal venous insufficiency in these patients did not reveal the clinical efficacy of endovascular procedures [18,19,20,21,22,23]. However, these treatments focused on the pathological jugular valves and primarily comprised balloon angioplasty and/or stenting for such aberrant valves. These approaches were not necessarily correct. Indeed, an expanded analysis of results of the BRAVE DREAMS (BRAin VEnous DRainage Exploited Against Multiple Sclerosis) trial revealed that only a subgroup of multiple sclerosis patients benefited from endovascular angioplasty at the level of jugular valve; it comprised patients presenting with horizontal endoluminal defects of the IJV and segmental stenosis with short endoluminal lesions of this vein [24].

In our previous paper [25], based on the results of flow simulations, we suggested that strictures located at the level of the jugular foramen are probably more clinically relevant than the pathological jugular valves. These computer simulations suggested that strictures in the upper part of IJV are probably more clinically relevant. Importantly, there are some clinical reports on such entrapment of IJV by bony abnormalities in the upper part of this vein. It has also been demonstrated that correction of these strictures can decrease neurological symptoms [26,27,28].

This in silico study is the continuation of that research [25] and was aimed at understanding the influence of stenoses located at the beginning of IJV on the flow through this vein and the functioning of the jugular valve located downstream.

## 2. Materials and Methods

For the purpose of this study computational fluid dynamics (CFD) software, the COMSOL Multiphysics, version 5.1 (COMSOL Inc., Burlington, MA, USA), was used. For numerical simulation of the real-time blood flow in normal and abnormal IJVs, two-dimensional (2D) models of IJVs were built. The 2D computational domain, which was 200 mm long (L) and 15 mm wide (D) [29], involved stenosis at its beginning and a hyperelastic venous valve downstream (Figure 2). Figure 2A particularly shows the nomenclature of IJV 2D model, where the input and output of IJV are represented by 1 and 2, whereas the flexible venous valve leaflets are denoted by 3 and 4. The fluid domain is represented by 5, whereas the walls of IJV are considered as a no-slip boundary shown by 6. A proximal stenosis (d_1_) and gap between valve leaflets (d_2_) are shown in Figure 2B. The geometry is discretized into free triangular components by considering all quality measures, which include optimal skewness, orthogonality, and aspect ratio (Figure 2C).

The grid independency study was conducted for the maximum velocity in the domain at *t* = 3 s. The study was conducted in a stepwise fashion, starting with a coarser mesh and then reducing the mesh element size until the maximum velocity of the blood in IJV became constant. Figure 3 shows the mesh independence study graph in which the mesh domain was optimized for 15,832 triangular elements.

This model had a symmetric 2-leaflets valve. Models resembling veins with strictures in their upstream segments had a rapid contraction, which narrowed the lumen by 30%, 60%, or 75%. In total, four different 2D models of IJV were constructed: (A) vein without upstream stenosis; (B) vein with 30% upstream stenosis; (C) vein with 60% upstream stenosis; and (D) vein with 75% upstream stenosis (Figure 4).

Dirichlet boundary conditions (by specifying the values that a solution will take along the boundary of the domain like **u** = *U*_inlet_ and *p*_outlet_ = 0 Pa) were applied at the inlet and outlet boundaries of the modeled blood vessels. The whole domain was discretized into a further two subdomains, blood (Domain 1) and venous valve (Domain 2). The turbulent flow modeling was coupled with solid mechanics module using the fluid–structure interaction approach, as the Reynold number (Re) exceeds 2000 in all cases, where the constitutive relation for blood shear stress to shear strain was considered Newtonian. The nomenclature is described in Table 1.

The Reynolds Average Navier Stokes (RANS) *k*-*ω* numerical method was used to simulate the mean flow characteristics, as shown in Equations (1)–(4).
(1)$$\rho \left[\;\frac{\partial u}{\partial t}+\left(u\cdot \nabla \right)u\right]=\nabla \cdot \left(-pI+K:\mu \left[\nabla u+{\left(\nabla u\right)}^{T}\right]\right)+F$$
(2)ρ∇·u=0
(3)$$\rho \left[\frac{\partial k}{\partial t}+\left(u\cdot \nabla \right)k\right]=\nabla \cdot \left[\left(\mu +\mu_{T}\sigma^{*}\right)\nabla k\right]+P_{k}-\rho \beta_{0}^{*}\omega k$$
(4)$$\rho \left[\frac{\partial \omega }{\partial t}+\left(u\cdot \nabla \right)\omega \right]=\nabla \cdot \left[\left(\mu +\mu_{T}\sigma \right)\nabla \omega \right]+\alpha \frac{\omega }{k}P_{k}-\rho \beta_{0}\omega^{2}$$
where,
$$\begin{array}{c} K=\left(\mu +\mu_{T}\right)\left[\nabla u+{\left(\nabla u\right)}^{T}\right];\;\;\;\;\;\;\mu_{T}=\rho \frac{k}{\omega } \end{array}\;P_{k}=\mu_{T}\left[\nabla u:\left(\nabla u+{\left(\nabla u\right)}^{T}\right)-\frac{2}{3}{\left(\nabla \cdot u\right)}^{2}\right]-\frac{2}{3}\rho k\nabla \cdot u\;\begin{array}{cccc} \alpha =\frac{13}{25}; & \beta =\beta_{0}f_{\beta }; & \beta^{*}=\beta_{0}^{*}f_{\beta }; & \sigma =\sigma^{*}=\frac{1}{2} \end{array}\;\begin{array}{ccc} \beta_{0}=\frac{13}{125}; & f_{\beta }=\frac{1+70\chi_{\omega }}{1+80\chi_{\omega }}; & \chi_{\omega }=\left|\frac{\Omega_{ij}\Omega_{jk}S_{ki}}{{\left(\beta_{0}^{*}\omega \right)}^{3}}\right| \end{array}\;\begin{array}{c} \beta_{0}^{*}=\frac{9}{10};\;\;\;\;\;\;f_{\beta }=\left\{\begin{array}{c} \begin{array}{cc} 1 & \chi_{k}\leq 0\\ \frac{1+680\chi_{k}^{2}}{1+400\chi_{k}^{2}} & \chi_{k}>0 \end{array}\chi_{k}=\frac{1}{\omega^{3}}\left(\nabla k\cdot \nabla \omega \right) \end{array}\right. \end{array}\;\begin{array}{cc} \Omega_{ij}=\frac{1}{2}\left(\frac{\partial {\bar{u}}_{i}}{\partial x_{j}}-\frac{\partial {\bar{u}}_{j}}{\partial x_{i}}\right); & S_{ij}=\frac{1}{2}\left(\frac{\partial {\bar{u}}_{i}}{\partial x_{j}}+\frac{\partial {\bar{u}}_{j}}{\partial x_{i}}\right) \end{array}\;$$

The incompressible flow without considering inertial terms (Stokes flow) was modeled. The material of the venous valve (Domain 2) was considered to be hyperelastic (a material whose constitutive model of stress–strain relationship derives from a strain energy density function rather than Hooke’s law). The neo-Hookean was used with compressible material features, and the mathematical model is given below in Equations (5) and (6).
(5)$$\rho_{s}\frac{\partial^{2}u_{d}}{\partial t^{2}}=\nabla {\left(FS\right)}^{T}+F_{V}$$
(6)$$\sigma =J^{-1}\;FSF^{T}$$
where,
$$\begin{array}{cc} F=I+\nabla u_{d}; & S=S_{ext}+\left(\frac{\partial W_{s}}{\partial \epsilon }\right) \end{array}$$
$$\begin{array}{cc} \;\;\;\;\;\;\;\;\;\;\;\;\;\;\;\;\;\;W_{s}=\frac{1}{2}\mu \left(I_{1}-3\right)-\mu \ln\left(J_{\mathrm{el}}\right)+\frac{1}{2}\;\lambda {\left[\ln\left(J_{\mathrm{el}}\right)\right]}^{2}; & \epsilon =\frac{1}{2}\left(F^{T}F-I\right) \end{array}$$
$$\begin{array}{ccc} J=\det\left(F\right); & \mu =\frac{E}{2\left(1+\nu \right)}; & \lambda =\frac{E\cdot \nu }{\left(1+\nu \right)\left(1-2\nu \right)} \end{array}\;$$

Lamé’s parameters for hyperelastic are *μ* = 1.0 MPa and *λ* = 1.5 MPa, considering modulus of elasticity *E* = 2.6 MPa and Poisson’s ratio *ν* = 0.3 [30], whereas the properties of the fluid were set as follows: the density of fluid 1055 kg m^−3^ (density of blood at 37 °C [31,32]) and the dynamic viscosity of fluid 2.78 × 10^−3^ Pa·s (viscosity of blood at 37 °C). The flow velocity was 16 ± 4 cm/s, which is a typical IJV velocity in humans in the supine body position [33]. The density of the valve leaflets was set at 1200 kg m^−3^ [34]. To simulate fluctuations of flow velocity in the modeled vein in a living subject resulting from the pulsation of the adjacent carotid artery and respiratory movements, the sine function of velocity at the inlet (Boundary 1) of the model was applied (Figure 2A). This sine function plotted the period of fluid velocity against time, as shown in Equation (7). The initial velocity of the fluid was 12 cm/s, which increased up to 16 cm/s (mean velocity) at *t* = 1.5 s and finally reached 20 cm/s (maximum velocity of blood) at *t* = 3 s, whereas it dropped back to 12 cm/s at *t* = 6 s.
(7)$$U_{inlet}=\left[8\;\sin\left(\;\frac{t}{6}\pi \right)+12\right]\;\;\mathrm{cm}/s$$

The static pressure boundary condition was used at the outlet (Boundary 2) of the modeled blood vessel as shown in Figure 2A, and the pressure value was set at *p* = 0 Pa. For the venous valve, both ends were fixed at the external wall of the model using *U*_solid_ = 0 cm/s condition, while the rest of the valve was allowed to move freely within the fluid regime. All simulations were continued until 6 s and the graphical representations of the flow were analyzed. All the computations were executed in the Intel-INSPIRON (Intel, Santa Clara, CF, USA) equipped with the Intel-R Core-TM i7-11 Gen processor and the Intel-R Iris-XR Plus graphic card, and it took around 11 h of computational time for one case.

## 3. Results

In the model of vein without upstream stenosis and a symmetric 2-leaflets valve, the flow was undisturbed (Figure 4A, Figure 5A and Figure 6A). Valve leaflets opened and closed symmetrically due to the vortices developing downstream of the valve [35]. A similar flow pattern was seen in the model with minor 30% stenosis and a normal 2-leaflets valve. In this model, there were vortices located downstream of the stenosis, but they did not significantly disturb the outflow, and neither did these vortices influence the functioning of valve leaflets (Figure 4B, Figure 5B and Figure 6B).

In the case of more severe upstream stenosis, with 60% and 75% narrowing of the lumen, by contrast to the above-discussed models in which the centerline velocity was positioned centrally, here the centerline velocity was positioned asymmetrically and the areas of reverse flow and flow separation developed (Figure 4C,D, Figure 5C,D and Figure 6C,D).

Importantly, in the models with significant upstream stenoses, vortices making the flow highly asymmetric were present at the entire course of the models, and not only just downstream of the stenosis, as in the model without significant upstream narrowing. These asymmetric flow pattern resulted in asymmetric bending of elastic valve leaflets (Figure 7).

We also studied the relationship between the maximum velocities observed in the models of blood vessel depending on the degree of upstream stenosis. Graphical representation of the results is shown in Figure 8. The humps in flow velocity curve were observed at the stenosis and venous valve position as shown in this figure. The blood velocity increased significantly at stenosis for 60% and 75% cases, while there was only a minor increase of the velocity in the model with 30% stenosis. Interestingly, in two models with major stenosis (60% and 75%), velocity decreased just behind the stricture, which was not seen in the model with 30% stenosis. This phenomenon was probably related to flow separation in the cases with the major upstream stenoses.

## 4. Discussion

This study confirmed the results of our previous study [25], which was based on a different computational fluid mechanics (CFM) package. Similar to the findings of this research, we demonstrated that stenosis located at the beginning of the internal jugular vein can significantly affect the blood flow pattern. However, the most important finding is the phenomenon of valve leaflets distortion by asymmetric vortices evoked by nozzle-like strictures located upstream. To the best of our knowledge, this is the first in silico study on the investigation of the functioning of the jugular veins in the settings of IJV abnormal morphology. This was not possible to demonstrate on a previous CFM software, since the Flowsquare+ that was used in this study does not provide a possibility of building flexible structures.

These findings highlight the possible pathogenesis of abnormal jugular valves. Atypical structured jugular valves can be found in some healthy individuals, but these are primarily present in patients suffering from neuroinflammatory and neurodegenerative diseases [36,37,38]. Although some clinicians claimed a causative role for such aberrant jugular valves in neurological disorders, such an association has not been unequivocally proved [39]. The current consensus is that a majority of abnormal jugular valves represent congenital pathology, the so-called truncular malformations [40,41]. Still, research on such valves in multiple sclerosis patients suggested that their abnormality worsens with time [39]. Until now, the mechanism of such a progression has remained unclear. Our numerical simulation of blood flow in the IJV that exhibits significant stenosis in its upper segment offers a possible explanation: asymmetric vortices arising in the upper part of this vein, which are evoked by stenosis at the level of the jugular foramen (e.g., caused by an elongated transverse process of the first vertebra), distort leaflets of the jugular valve. The pathological flow pattern around such distorted leaflets can theoretically alter the leaflets’ physiology, leading to even more abnormal geometry. Indeed, jugular valves histopathological studies in multiple sclerosis patients revealed their abnormal ultrastructure, including defective endothelium and an inverted ratio between type I and type III collagen [42,43].

Our current work suggests that from a physical point of view there is an interplay between pathologies of IJV localized in the upper and lower part. To achieve a satisfactory clinical outcome in the patients, both types of lesions should be properly diagnosed and addressed. For example, stenosis at the beginning of the IJV would require resection of abnormal bony structures, while pathological jugular valves would require venous angioplasty. This can explain the low clinical efficacy of angioplasty procedures for the treatment of multiple sclerosis [23]. Of course, our findings should be confirmed by more precise investigations, including studies on real patients. Still, our report can provide a useful framework for future clinical surveys.

We acknowledge that there are limitations to our study. The findings of this research, albeit inspiring, should be ascertained using 3D CFD software. We utilized 2D models of the COMSOL Multiphysics software and considered the cylindrical plane to visualize the velocity profile in the modeled blood vessel. It should be mentioned that 2D simulations are widely used in the engineering sciences and are regarded to be within an order of magnitude of accuracy, as long as the problem is well-defined in the software and care is taken with regards to appropriate meshing and boundary conditions [44,45]. Still, there are known limitations of such a 2D approach. One of the solutions to this problem is the use of 2D-axisymmetric geometry, which limits these shortcomings, while very large computing resources and computing time, as in the case of 3D modeling, are not required [46,47]. This method should probably be used in future studies on the flow in the models of the IJV.

Additionally, morphology-based IJV models instead of simplified models would provide more detailed insight into the altered blood flow phenomena. Potentially, other properties of the IJV and surrounding tissues could contribute to abnormal outflow through this blood vessel. Besides, we studied one IJV blood flow, although in the living subjects blood flows out from the brain through two IJVs, as well as through the vertebral venous plexuses located on both sides of the spinal column. More precise and adequate modeling should consider these alternative outflow routes. It should be emphasized that a steady state was not achieved in our simulations due to the required computational expense. However, the computational results at 6 s provide a unique insight into the flow effects of strictures in the IJV. Finally, it should be mentioned that in this study the fluid was considered Newtonian. Blood is a non-Newtonian fluid. Since blood is shear-thinning fluid, slowing down its flow is associated with a higher flow resistance than that of a Newtonian fluid.

## 5. Conclusions

We demonstrated that our working hypothesis is credible and that impaired outflow from the brain through the internal jugular vein is likely to be primarily caused by pathological strictures in the upper part of these veins. In addition, jugular valve pathology can be exacerbated by flow disturbances evoked by strictures in the upper segments of these veins.

## Figures and Tables

**Figure 1 diagnostics-12-02862-f001:**
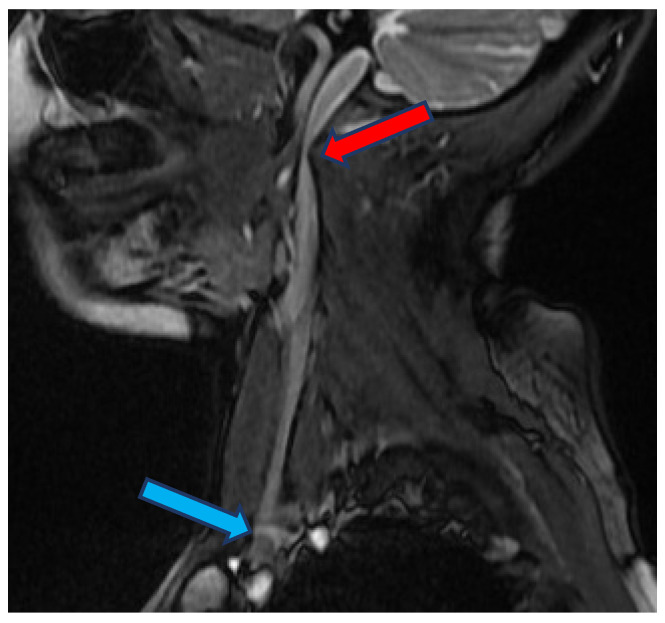
The internal jugular vein with stricture in its upper part (red arrow), blue arrow points to the location of the jugular valve.

**Figure 2 diagnostics-12-02862-f002:**
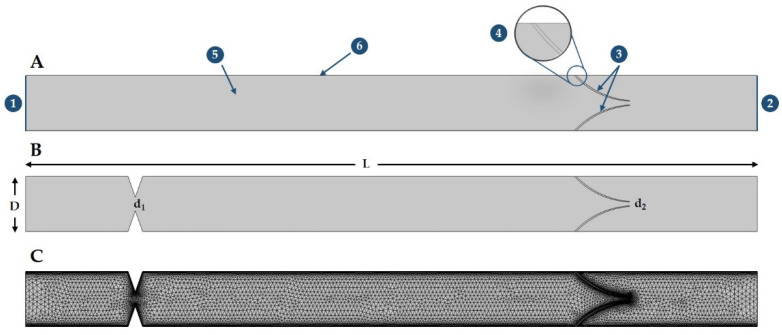
The scheme of the 2-dimensional model of the internal jugular vein and symmetrical 2-leaflets hyperelastic valve downstream; flow is from left to the right; this idealized model resembles real internal jugular vein with tandem stenosis from Figure 1. (**A**) without stenosis; (**B**) with 60% rigid stenosis at the beginning of this blood vessel (d_1_) and flexible valve (d_2_) downstream; and (**C**) free-triangular mesh with boundary layer.

**Figure 3 diagnostics-12-02862-f003:**
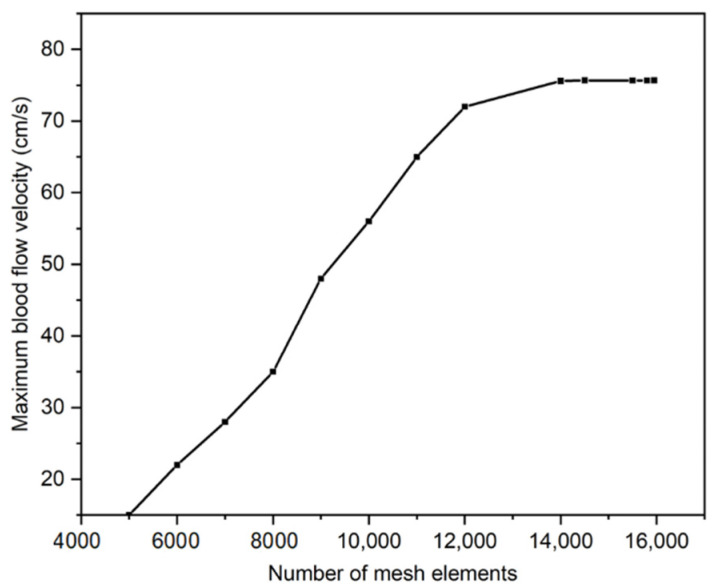
Mesh independence study: maximum blood flow velocity against the number of mesh elements in the domain.

**Figure 4 diagnostics-12-02862-f004:**
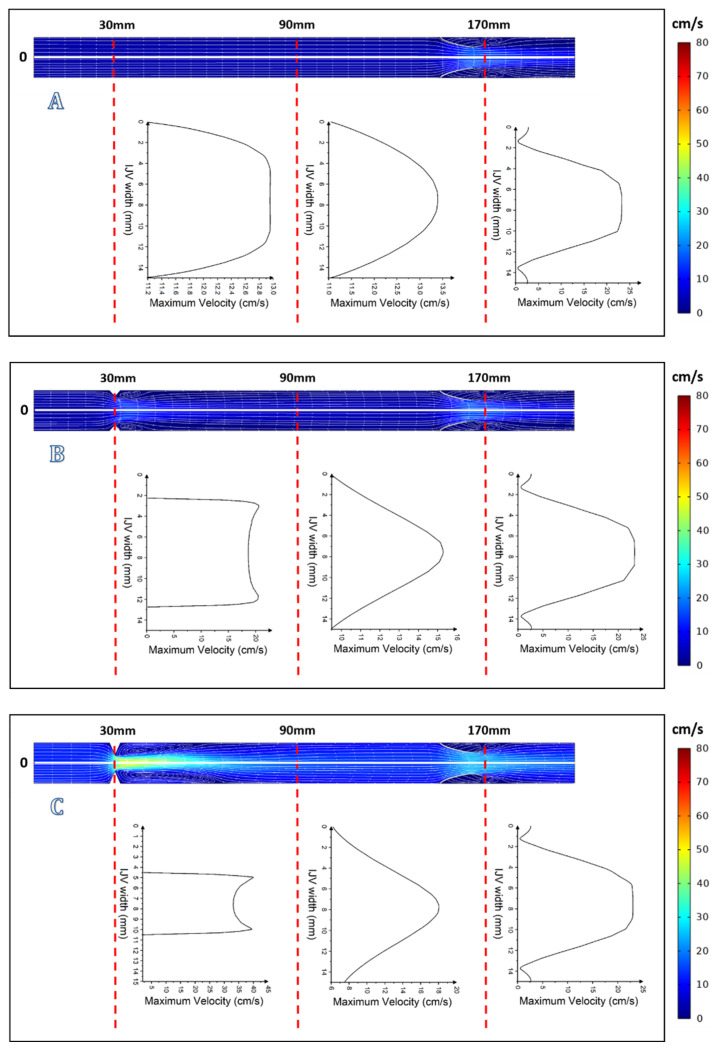
Models of the internal jugular vein and symmetric 2-leaflets valve with velocity profiles at *t* = 0.25 s: (**A**) without upstream stenosis; (**B**) with 30% upstream stenosis; (**C**) with 60% upstream stenosis; and (**D**) with 75% upstream stenosis.

**Figure 5 diagnostics-12-02862-f005:**
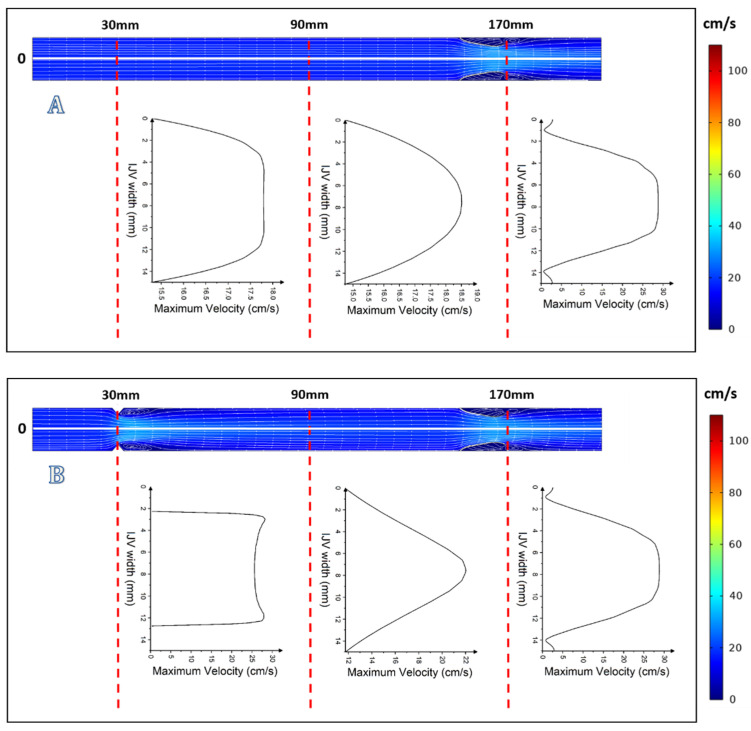
Graphical representation of flow with velocity profiles at *t* = 1.5 s: (**A**) without upstream stenosis; (**B**) with 30% upstream stenosis; (**C**) with 60% upstream stenosis; and (**D**) with 75% upstream stenosis.

**Figure 6 diagnostics-12-02862-f006:**
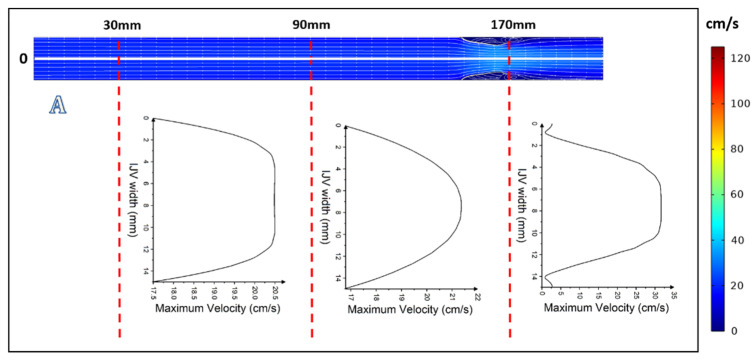
Graphical representation of flow with velocity profiles at *t* = 3 s: (**A**) without upstream stenosis; (**B**) with 30% upstream stenosis; (**C**) with 60% upstream stenosis; and (**D**) with 75% upstream stenosis.

**Figure 7 diagnostics-12-02862-f007:**
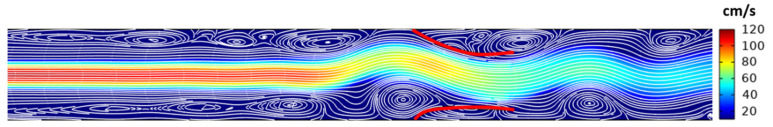
Asymmetric bending of valve leaflets (red) by vortices in the model D (with 75% upstream stenosis).

**Figure 8 diagnostics-12-02862-f008:**
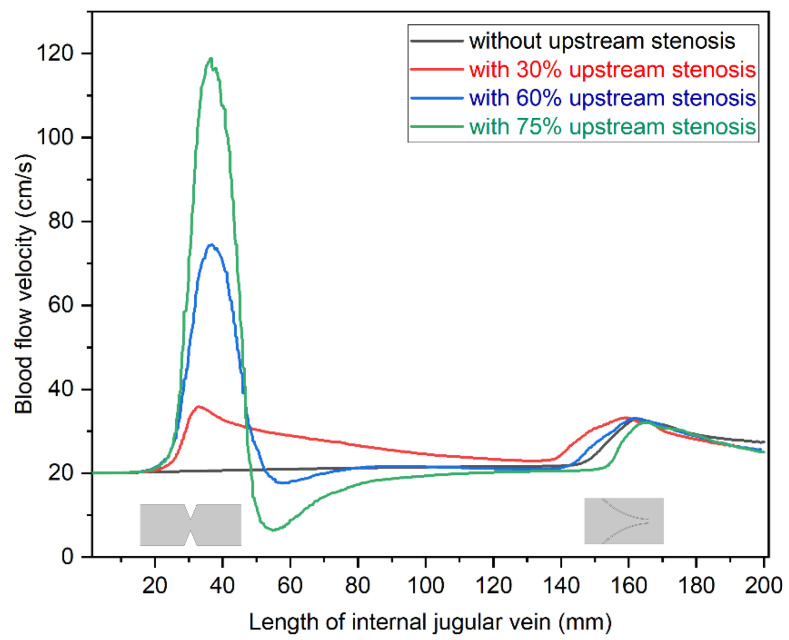
Comparison of maximum velocity (20 cm/s) profiles against the length of the modeled internal jugular vein, recorded at *t* = 3 s.

**Table 1 diagnostics-12-02862-t001:** Nomenclature.

Quantity (Symbol)	Unit	Quantity (Symbol)	Unit
Density of blood (*ρ*)	kg m^−3^	Partial derivative $\left(\partial \right)$	–
Time (*t*)	s	Fluid velocity (**u**)	m s^−1^
Fluid pressure (*p*)	Pa	Turbulent intensity (**I**)	–
Dynamic viscosity (*μ*)	Pa s	External forces (**F**)	N
Turbulent kinetic energy (*k*)	J	Turbulent viscosity (*μ_T_*)	kg^2^ s m^−3^
Specific dissipation rate (*ω*)	J kg^−1^ s^−1^	Mean rotation-rate tensor (*Ω_ij_*)	–
Mean strain-rate tensor (*S_ij_*)	–	Venous valve density ($\rho_{s}$)	kg m^−3^
Displacement vector (**u**_d_)	m	Deformation gradient (**F**)	–
Stress tensor (**S**, *σ*)	N m^−2^	Volume force (**F***_V_*)	N m^−3^
Elastic volume ratio (*J_el_*)	–	Identity tensor (**I**)	–
External stress tensor (**S***_ext_*)	N m^−2^	Strain energy density (*W_s_*)	J
Lamé’s second parameter (*μ*)	MPa	Lamé’s first parameter (*λ*)	MPa
Modulus of elasticity (*E*)	MPa	Poisson’s ratio (*ν*)	–

## Data Availability

Metadata containing all graphic files depicting the flow in the modeled veins can be found in the repository: https://repod.icm.edu.pl/dataset.xhtml?persistentId=doi:10.18150/LFNDPV (accessed on 21 June 2022). Preprint version of this paper is available in SSRN depository at: https://papers.ssrn.com/sol3/papers.cfm?abstract_id=4143101 (accessed on 22 June 2022).

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
