# Peer review of "Results of Numerical Modeling of Blood Flow in the Internal Jugular Vein Exhibiting Different Types of Strictures"

_diagnostics, 2022, doi:10.3390/diagnostics12112862_

Round 1
Reviewer 1 Report
The paper is presented to a very good standard of English with good quality images and figures.
The paper presents results from two-dimensional (2D) CFD simulations and in this respect I have serious doubts regarding the relevance of the results and the conclusions made in the paper. A 2D simulation implies that the third dimension is infinite, the results reported were essentially obtained for a (rectangular) duct with infinite depth – this does not represent the actual physical geometry.
The authors recognise this and state “We acknowledge that there are limitations to our study. The findings of this research, albeit inspiring, should be ascertained using 3D CFD software”.
I suggest that the results for true 3D simulations will behave in a considerably different manner to those reported. As a consequence I do not believe the current results should be published.
In terms of the paper, there are considerable omissions or misunderstandings regarding details of the CFD simulations themselves.
Line 109
“Dirichlet boundary conditions were applied at the inlet and outlet bounda-109 ries of the modeled blood vessels”
What is a Dirichlet boundary condition, and what boundary conditions (velocity, pressure,symmetry etc.) were applied.
Line 112
“where the constitutive relation for blood shear 113 stress to shear strain was considered Newtonian”
The choice of a Newtonian fluid should be justified. The authors later acknowledge this is not the case – line 235
“Finally, it should be mentioned that in 235 this study the fluid was considered Newtonian. Actually, blood is a 236 non-Newtonian fluid. Since blood is shear-thinning fluid, slowing down its flow 237 is associated with a higher flow resistance than that of a Newtonian fluid.”
Line 114
“A RANS k-ω numerical method”
What is RANS and what is k-ω ? No justification is provided as to why a turbulent flow is being simulated – this should be justified by reference to Reynolds number of the flow.
The authors provide equations, but no nomenclature is provided. As a minimum appropriate citations should be provided.
Equation (1) must be incorrect since no time derivative is shown, yet the simulations were transient.
The authors should justify how they can use a RANS turbulence module for transient flow.
Line 118
“hyper-elastic material. The Neo-Hookean was used with compressible material features”
What is a hyper-elastic material ? What is Neo-Hookean. No citation provided, which again suggests that the authors are using built-in models within COMSOL, without understanding what they represent.
Again, no nomenclature is provided for the equations.
Line 131
A sinusoidal boundary condition was implemented for the inlet velocity, but what velocity profile was adopted across the duct (uniform) ? The choice should be justified.
No boundary condition information is provided for k and ω (omega). This must be provided and justified.
Line 140
The authors state “In the model of vein without upstream stenosis and a symmetric 2-leaflet 140 valve, the flow was undisturbed and laminar”
The simulation results are not laminar by definition, since a RANS turbulence model was employed. The phrase should be rewritten.
Author Response
Response to the remarks of the Reviewer 1:
Please see attachment

Reviewer 2 Report
The study submitted presents a computational analysis of fluid flows within models that mimic the internal jugular artery with or without constrictions. To be publishable the study has to address quite a few issues:
- The title of the paper is too long. It should be more concise, not an abstract of the paper.
- While Figure 2, gives a sketch of the geometry studied, no actual dimensions are given.
- The model is 2D. Given the direct reference to the internal jugular artery, for the model to have relevance to the specified problem the model should be simulated in 3D.
- The assumption of the model is that the blood can be treated as a Newtonian fluid. However, blood is a complex fluid that exhibits non-Newtonian rheological characteristics. There is no reason for those to be ignored in particular considering the drastic constrictions, and thus large velocity gradients, used in the model. Some of these limitations are acknowledged by the authors, but models for the blood dynamic viscosity do exists and have been implemented in the software used, i.e. COMSOL, as well.
- No justification is given for the Lame parameters used in the elastic model for the valve.
- Is steady state achieved in the model? The simulation seems to be run for only one pulsation.
- Figure 6 and Figure 7 – explain why asymmetric flows immediately after the constriction are observed in one system and not the other. For Figure 6, provide snapshots of the flow field at different timestamps like in Figure 7.
- How is this study different than the one from reference [13]. Just changing the software package is not justification enough for a publication.
- A large portion of the “Discussion” from Page 11, would be better suited for the introduction, as it justifies the clinical applications of a computational study on this particular system.
- The study could be strengthened by a simulation on a realistic geometry for the internal jugular valve. Such geometry can be extracted nowadays from MRI images.
Author Response
October 31th, 2022
Editors of the Diagnostics
Dear Madam/Sir
Herein is enclosed a revised version of the manuscript titled: “Results of Numerical Modeling of Blood Flow in the Internal Jugular Veins Exhibiting Different Types of Strictures” – previously: “Stenosis at the Beginning of the Internal Jugular Vein Compromises Outflow through this Blood Vessel and Can Alter the Geometry of the Jugular Valve Located Downstream: Results of Numerical Modeling of Blood Flow”.
Amended parts of the text are underlined in red.
Specifically, we re-simulated the flow in the models following remarks and suggestions of the Reviewers. Since with new parameters and corrected equations it was not possible to simulate in a reasonable time (the simulation would require several days of computer work) the flow in the case of vein with septum, we decided to perform simulations in three, instead of two, model with proximal rigid stricture. Details are given in the Materials and Method chapter:
“This model had symmetric 2-leaflets valve. Models resembling veins with strictures in their upstream segments had a rapid contraction, which narrowed the lumen by 30%, 60% or 75%. In total, 4 different 2D models of IJV were constructed: (A) vein without upstream stenosis; (B) vein with 30% upstream stenosis; (C) vein with 60% upstream stenosis; and (D) vein with 75% upstream stenosis (Figure 4).”
Consequently, figure legends (Figs. 4-6) are also amended.
Response to the remarks and proposals of the Reviewer 2:
We kindly thank you for valuable comments regarding our submission.
- Following your suggestion, we changed the paper title to: “Results of Numerical Modeling of Blood Flow in the Internal Jugular Veins Exhibiting Different Types of Strictures”
- We provided dimensions of the models depicted in the Figure 2 in the Materials and Methods section:
“The 2D computational domain, which was 200 mm long (L) and 15 mm wide (D), involved stenosis at its beginning and a hyperelastic venous valve downstream (Figure 2).”
- The problem associated with 3D modelling has already been discussed in the reply No 1 to the Reviewer 1 (see: above).
- This issue has also been discussed in these comments – see reply No 3 to the Reviewer 1.
- We assumed that the structural properties of the valve behaved like a compressible hyperelastic material. To cater that behavior, neo-Hookean constitutive model was used to implement the hyperelastic behavior of the valve. We gave the parapeters used in the Materials and Methods section:
“Lamé's parameters for hyperelastic are λ = 1.5 MPa and μ = 1.0 MPa, considering modulus of elasticity E = 2.6 MPa and Poisson’s ratio ν = 0.3 whereas the properties of the fluid were set as follows: the density of fluid 1055 kg m-3 and the dynamic viscosity of fluid 2.78×10-3 Pa·s at 37 °C”
Parameters that we used to model hyperelastic material of the valve were similar to those used by Gataulin and colleagues who studied the level of blood stagnation in the area of venous valve depending on elasticity of such a valve (doi:10.1088/1742-6596/1359/1/012010).
- In our study the steady state was not achieved. This was because it would require a lot of computational time and cost, which was not available at the moment. Still, the simulation that run for only 6 s provided us with enough interesting data. We gave a comment on this issue in the Discussion chapter:
“It should also be emphasized that in our simulations the steady states were not achieved. This was because it would require a lot of computational time and cost, which was not available at this stage of research. Still, the simulations that run for 6 s provided us with enough interesting data.”
- The phenomenon of asymmetric flow pattern downstream of a nozzle-like stricture of a tube is well known in the engineering. Such a behaviour of flow, however, can be seen only if the nozzle-like stenosis is significant. A small stenosis does not affect significantly the flow. Asymmetry of the flow depends on percentage of constriction opening, the smaller the size of opening, the more asymmetric the flow becomes after the constriction. It is in agreement with diffraction of fluid wave. There is a number of papers describing asymmetric flows after such a major stenosis (in addition to engineering works there are also such papers on the flow distally from stenoses of the carotid arteries). Our team has already published the paper that revealed asymmetric flow in a case of severe stenosis of the internal jugular vein, but not if such a stenosis was minor (doi:10.1177/0268355521996087; Refrence No 24).
- In the previous research it was not possible to model flexible valves. Model with flexible valves (such as they are in real living subjects) provided us with an insight into the behaviour of these valves in the settings of atypical flow patterns. We gave a comment on that in the Discussion section:
“This was not possible to demonstrate on a previous CFM software, since the Flowsquare+ that was used in this study does not provide a possibility of building flexible structures.”
- Following suggestion of the Reviewer, we moved part of the Discussion to the Introduction chapter.
- We appreciate your suggestion that a simulation on a realistic geometry of the jugular valve could be scientifically valuable. Still, our study was primarily aimed at investigating what happens in a vein with symmetric geometry if there is a significant stenosis in the upper segment of this vein.
Besides, even if an extraction of actual geometry of the jugular valve from MR scans was possible (actually, it would require scans made by 7T scanners, with very sophisticated imaging protocols, considering tiny structure of such valves; invasive catheter angiography imaging seems to be better suited for such a purpose), there will be problems with 3D modelling (see reply No 1 to the Reviewer 1) and such a modelling is currently beyond financial and computational capabilities of our team.
Still, added a commentary of potential importance of other than the valve itself parameters of the internal jugular vein and also surrounding tissues, which should be investigated in the future:
“Also, morphology-based IJV models instead of simplified models would provide more detailed insight into the altered blood flow phenomena. Potentially, other properties of the IJV and surrounding tissues could contribute to abnormal outflow through this blood vessel.”
Reviewer 3 Report
The authors present a computational study evaluating the effects of nozzle-like strictures/stenoses on flow through internal jugular veins (IJV). The computational fluid dynamics modeling suggest that severe stenosis in the upper part of the IJV causes flow disturbances and asymmetric velocity profiles downstream towards the jugular valves. The authors suggest the disturbed flow caused by these strictures may exacerbate jugular valve pathology. The results are very interesting and provide further impetus for understanding how IJV flow pathology may lead to therapies for neurodegenerative disorders. I have several suggestions to help the authors improve their manuscript:
Comments:
1. Introduction (p. 2, line 53): The previous paper should be cited at first mention. Is this the same as the one in Reference 13?
2. Materials and Methods (p.2, line 67): Are the idealized 2D models (represented by Figure 2), with concentric stenosis, representative of the 3D structural/flow domain variations that may exist in this geometry? For example, in Figure 1, the stenosis (red arrow) appears to be eccentric, while the diameters of the vein and jugular valve (blue arrow) are not consistent along the length of the vein. Is there a reason for using 2D models rather than 3D models used previously (i.e., in Ref. 13 – Simka and Latacz, Phlebology, 2021)?
3. Materials and Methods (p. 3, line 76): A “normal” IJV scheme/mesh should be provided for comparison.
4. Materials and Methods (p. 3, line 99): Figure 4 should show details for each of the features described in this paragraph, particularly the stenoses and leaflets, since these are too small in the current Figure 4. Furthermore, the flow behaviors seen in these figures seem to be a bit unusual at t=0.25 s, particularly the concave (“U-shaped”) peaks for the velocities at the stenoses in panels B-D, and valves in panels B-C. Is the flow not fully developed, or is there another reason for this behavior?
5. Materials and Methods (p. 5, lines 121-123): Blood density and dynamic viscosity should ideally be provided for the same temperature, 37°C. Is there a reason for using the density at room temperature?
6. Materials and Methods (p. 6, line 136): What was the temporal resolution of the simulation? Please discuss the selection of the observation time t=5 s in Figures 5-6, as peak fluid velocity is at t=3 s (as mentioned in lines 130-131).
7. Results (p. 10, line 196): The caption for Figure 8 should specify to which of the case studies this detail refers.
8. Results (p. 10, line 196) and Discussion: The authors should describe the relationship between the maximum velocities observed in the stenosed vessels with 2-leaflet and 1 septum-like valve as compared to those observed in normal IJVs. An analysis quantifying the effect of geometrical variation (i.e. stenosis size, type of valve) on the flow outcomes (i.e. maximum velocity, stresses, etc.) as compared to normal geometry would be appropriate.
9. Discussion (p. 10, lines 227-238): The limitations should be moved to the end, just before the conclusion. Also, can the authors discuss the potential effect of blood vessel properties (i.e., stiffness, motion, patient-specific geometry) on the flow outcomes, in addition to the hyperelastic valves included in these simulations?
10. Discussion (p. 11, lines 258-262): The authors suggest that abnormal flow conditions due to strictures in the upper part of the IJV play a larger clinical role in neurological pathologies than aberrant jugular valves. Can the authors quantify or compare the effects of the two based on the results of this study?
Author Response
October 31th, 2022
Editors of the Diagnostics
Dear Madam/Sir
Herein is enclosed a revised version of the manuscript titled: “Results of Numerical Modeling of Blood Flow in the Internal Jugular Veins Exhibiting Different Types of Strictures” – previously: “Stenosis at the Beginning of the Internal Jugular Vein Compromises Outflow through this Blood Vessel and Can Alter the Geometry of the Jugular Valve Located Downstream: Results of Numerical Modeling of Blood Flow”.
Amended parts of the text are underlined in red.
Specifically, we re-simulated the flow in the models following remarks and suggestions of the Reviewers. Since with new parameters and corrected equations it was not possible to simulate in a reasonable time (the simulation would require several days of computer work) the flow in the case of vein with septum, we decided to perform simulations in three, instead of two, model with proximal rigid stricture. Details are given in the Materials and Method chapter:
“This model had symmetric 2-leaflets valve. Models resembling veins with strictures in their upstream segments had a rapid contraction, which narrowed the lumen by 30%, 60% or 75%. In total, 4 different 2D models of IJV were constructed: (A) vein without upstream stenosis; (B) vein with 30% upstream stenosis; (C) vein with 60% upstream stenosis; and (D) vein with 75% upstream stenosis (Figure 4).”
Consequently, figure legends (Figs. 4-6) are also amended.
Response to the remarks and proposals of the Reviewer 3:
We kindly thank you for valuable comments regarding our submission.
- Following the suggestions of the Reviewer, we moved the citation No 24 (previously 13) to avoid misunderstanding.
“In our previous paper [24], based on the results of flow simulations, we suggested that strictures located at the level of the jugular foramen are probably more clinically relevant than the pathological jugular valves”.
- Rationale of using 2D instead of 3D modelling has already been thoroughly discussed in the reply No 1 to the Reviewer 1. In brief, 3D simulations require much more (by a level of magnitude) computational time and power, which was beyond or capabilities. In this study each case required about 11 h of computational work; 3D modelling would probably need several weeks if a powerful computer unit (those are very expensive to use) were not used.
In this particular case the geometry of studied blood vessel was cylindrical. Therefore, only mid-cylinder plane (plane of symmetry) was simulated to capture maximum variations in the physical parameters. The physical parameters do not vary in circumferential planes and therefore it can be assumed to be 2D, in order to avoid the computational complexity without losing the crucial results. Such an approach is widel used in the engineering.
- Following suggestion of the Reviewer, we added a schematic representation of the vein with tandem stenosis, and changed the legend of the Figure 2, in order to give the Readers reference of this model to the real vein shown in Figure 1.
“Figure 2. The scheme of the 2-dimensional model of the internal jugular vein and symmetrical 2-leaflets hyperelastic valve downstream; flow is from left to the right; this idealized model resembles real internal jugular vein with tandem stenosis from the Figure 1. (A) without stenosis; (B) 60% rigid stenosis at the beginning of this blood vessel (d1) and flexible valve (d2) downstream; and (C) free-triangular mesh with boundary layer.”
There is also a detailed description of the models in the Materials and Method section:
“Figure 2A particularly shows the nomenclature of IJV 2D model, where the input and output of IJV are represented by 1 and 2 whereas the flexible venous valve leaflets are denoted by 3 and 4. The fluid domain is represented by 5, whereas the walls of IJV are considered as a no-slip boundary shown by 6. A proximal stenosis (d1) and gap between valve leaflets are shown in Figure 2B. The geometry is discretized into free triangular components by considering all quality measures, which include optimal skewness, orthogonality, and aspect ratio (Figure 2C).”
- Following the suggestions of the Reviewer, we re-simulated the flow in the models. As can be seen (Figs.4-6) an awry shape of velocity profile at the level of stenosis becomes more evident as the degree of stenosis increases; in the model with minor stenosis there is almost a “normal” parabolic velocity profile. These “unusual” velocity profiles continue as asymmetric flows alongside the studied blood vessels. Our interpretation of these findings is that a nozzle-like stricture evokes flow separation and the so-called vena contracta phenomenon (the reason for this phenomenon is that fluid streamlines cannot abruptly change direction at the level of narrow stenosis). However, since we do not have enough evidence supporting this hypothesis, we decided not to comment on that in the current paper.
- Following suggestion of the Reviewer, we changed parameters of the fluid to those of 37o C, and performed all the calculations with this corrected viscosity.
Corresponding citations have also been changed accordingly.
“…the properties of the fluid were set as follows: the density of fluid 1055 kg m-3 and the dynamic viscosity of fluid 2.78×10-3 Pa·s at 37 °C”
- Following suggestions of the Reviewers, we amended the parameters of the fluid as well as the equations has been corrected. Consequently, new simulations were needed. In order to perform them correctly, with properly constructed mesh, we have done the independence study. Results of this analysis are shown in Figure 3.
“The grid independency study was conducted for the maximum velocity in the domain at t = 3 s. The study was conducted in a stepwise fashion starting with a coarser mesh and then reducing the mesh element size until the maximum velocity of the blood in IJV became constant. Figure 3 shows the mesh independence study graph in which the mesh domain was optimized for 15832 triangular elements.”
We acknowledge the fact that in our study the steady state was not achieved. This issue is already discussed in the reply No 6 to the Reviewer 2 (see: above).
- The caption for Figure 8 is corrected. Since some of the previous figures were changed, current number of this figure is 7.
“Figure 7. Asymmetric bending of valve leaflets by vortices in the model D (with 75% upstream stenosis).”
- Following suggestion of the Reviewer, we performed analysis of the relationship between the maximum velocities observed in the studied models of blood vessels. Results are shown in Figure 8, and presentation of these results is added:
“We also studied the relationship between the maximum velocities observed in the models of blood vessels depending on the degree of upstream stenosis. Graphic representation of the results is shown in Figure 8. The humps in flow velocity curve were observed at the stenosis and venous valve position as shown in this figure. The blood velocity increased significantly at stenosis for 60% and 75% cases, while there was only a minor increase of the velocity in the model with 30% stenosis. Interestingly, in two models with major stenosis (60% and 75%) velocity decreased just behind the stricture, which was not seen in the model with 30% stenosis. This phenomenon was probably related to flow separation in the cases with major upstream stenoses.”
The case with septum-like valve has been removed from revised version of this paper since computer simulations for this particular case, with amended parameters and corrected equations, would last at least several days.
- Following suggestion of the Reviewer, we moved part of the Discussion dedicated to the shortcomings, to the end of this chapter.
We also added a commentary of potential meaning of other properties of the vein studied:
“Potentially, other properties of the IJV and surrounding tissues could contribute to abnormal outflow through this blood vessel.”
- As we have written in the Discussion:
“However, these treatments focused on the pathological jugular valves and primarily comprised balloon angioplasty and/or stenting for such aberrant valves. Though, these approaches were not necessarily correct. Our computer simulations suggest that strictures in the upper part of IJV are probably more clinically relevant. There are some clinical reports on such entrapment of IJV by bony abnormalities in the upper part of this vein. It has also been demonstrated that correction of these strictures can decrease neurological symptoms”
a pathological role for the strictures in the upper segment of the internal jugular vein is only a suggestion. To validate this hypothesis a more complex computer model should be constructed, primarily comprising alternative outflow routes (such as vertebral veins). Yet, it should be the topic of future research.
Round 2
Reviewer 1 Report
I acknowledge the updated manuscript which reflects original review comments. However, I am afraid that I disagree with the authors response regarding the dimensionality of the simulations.
"We do not agree with the Reviewer’s assumption that 2D simulation implies that the third dimension is infinite. "
There is perhaps a misunderstanding on the part of the authors, and perhaps a limitation of COMSOL. A 2D-axisymmetric simulation should have been undertaken. The paper does not make any reference to 2D axisymmetric. A 2D simulation DOES exactly imply an infinite third dimension - that is the definition of 2D.
I would recommend that the authors find out more about 2D axisymmetric coordinates, and, as a minimum, add a statement in the paper to explain that these were not used.
The authors should consider what differences might occur between the 2D simulation and 2D-axisymmetric or 3D.
Author Response
We express gratitude for pointing out the limitations of the COMSOL package. We agree with the reviewer about the limitations of COMSOL, although we considered the cylindrical plane to visualize the velocity profile in the IJV. The authors further agree with the reviewer that the simulation should use 2D-axisymmetric geometry in COMSOL which will be considered in future simulations.
We added the commentary on this issue in the Discussion chapter and provided some references:
“We utilized 2D models of the COMSOL Multiphysics software, and considered the cylindrical plane to visualize the velocity profile in the modeled blood vessel. It should be mentioned that 2D simulations are widely used in the engineering sciences and are regarded to be within an order of magnitude of accuracy, as long as the problem is well defined in the software and care is taken with regards to appropriate meshing and boundary conditions [42,43]. Still, there are known limitations of such a 2D approach. One of the solutions to this problem is the use 2D-axisymmetric geometry, which limits these shortcomings, while a very large computing resources and computing time, as in the case of 3D modeling, are not required [44,45] This method should probably be used in future studies on the flow in the models of the IJV.”

Reviewer 2 Report
Most of the comments have been addressed successfully. The main items remaining are providing 3D simulations and modeling blood as a a non-Newtonian fluid. I do understand that these require more computational power, but that does not mean that they are impossible. It would be interesting to see if the results of 2D models correlate with those of 3D models.
Author Response
There is a commentary on 3D approach to the problem studied in the Discussion:
“The findings of this research, albeit inspiring, should be ascertained using 3D CFD software. We utilized 2D models of the COMSOL Multiphysics software, and considered the cylindrical plane to visualize the velocity profile in the modeled blood vessel. It should be mentioned that 2D simulations are widely used in the engineering sciences and are regarded to be within an order of magnitude of accuracy, as long as the problem is well defined in the software and care is taken with regards to appropriate meshing and boundary conditions [42,43]. Still, there are known limitations of such a 2D approach. One of the solutions to this problem is the use 2D-axisymmetric geometry, which limits these shortcomings, while a very large computing resources and computing time, as in the case of 3D modeling, are not required [44,45] This method should probably be used in future studies on the flow in the models of the IJV.”

Reviewer 3 Report
The authors present a significantly revised manuscript on their computational study evaluating the effects of nozzle-like strictures/stenoses on flow through internal jugular veins (IJV). The authors have satisfactorily addressed my comments, and I have a final minor comment to help improve the manuscript:
Comment:
1. Discussion (p. 13, lines 286-289): The comment about steady states not being achieved can be rephrased for brevity, i.e., “It should be emphasized that steady state was not achieved in our simulations due to the required computational expense. However, the computational results at 6 s provide a unique insight into the flow effects of strictures in the IJV.”
Author Response
We kindly thank you for valuable comments regarding our submission. We changed the fragment in the Discussion section following your suggestion.
“It should be emphasized that steady state was not achieved in our simulations due to the required computational expense. However, the computational results at 6 s provide a unique insight into the flow effects of strictures in the IJV.”
